# Towards a Neural Statistician

**Harrison Edwards**
School of Informatics
University of Edinburgh
Edinburgh, UK
`H.L.Edwards@sms.ed.ac.uk`

**Amos Storkey**
School of Informatics
University of Edinburgh
Edinburgh, UK
`A.Storkey@ed.ac.uk`

## Abstract

An *efficient* learner is one who reuses what they already know to tackle a new problem. For a *machine* learner, this means understanding the similarities amongst datasets. In order to do this, one must take seriously the idea of working with datasets, rather than datapoints, as the key objects to model. Towards this goal, we demonstrate an extension of a variational autoencoder that can learn a method for computing representations, or *statistics*, of datasets in an unsupervised fashion. The network is trained to produce statistics that encapsulate a generative model for each dataset. Hence the network enables efficient learning from new datasets for both unsupervised and supervised tasks. We show that we are able to learn statistics that can be used for: clustering datasets, transferring generative models to new datasets, selecting representative samples of datasets and classifying previously unseen classes. We refer to our model as a *neural statistician*, and by this we mean a neural network that can learn to compute summary statistics of datasets without supervision.

## 1 Introduction

The machine learning community is well-practised at learning representations of *data-points* and *sequences*. A middle-ground between these two is representing, or summarizing, *datasets* - unordered collections of vectors, such as photos of a particular person, recordings of a given speaker or a document as a bag-of-words. Where these sets take the form of i.i.d samples from some distribution, such summaries are called *statistics*. We explore the idea of using neural networks to learn statistics and we refer to our approach as a *neural statistician*.

The key result of our approach is a *statistic network* that takes as input a set of vectors and outputs a vector of summary statistics specifying a generative model of that set - a mean and variance specifying a Gaussian distribution in a latent space we term the context. The advantages of our approach are that it is:

- *Unsupervised*: It provides principled and unsupervised way to learn summary statistics as the output of a variational encoder of a generative model.

- *Data efficient*: If one has a large number of small but related datasets, modelling the datasets jointly enables us to gain statistical strength.

- *Parameter Efficient*: By using summary statistics instead of say categorical labellings of each dataset, we decouple the number of parameters of the model from the number of datasets.

- *Capable of few-shot learning*: If the datasets correspond to examples from different classes, *class embeddings* (summary statistics associated with examples from a class), allow us to handle new classes at test time.

## 2 Problem Statement

We are given datasets $D_i$ for $i \in \mathcal{I}$. Each dataset $D_i = \{x_1, \ldots, x_{k_i}\}$ consists of a number of i.i.d samples from an associated distribution $p_i$ over $\mathbb{R}^n$. The task can be split into learning and inference components. The learning component is to produce a generative model $\hat{p}_i$ for each dataset $D_i$. We assume there is a common underlying generative process $p$ such that $p_i = p(\cdot|c_i)$ for $c_i \in \mathbb{R}^l$ drawn

from $p(c)$. We refer to $c$ as the *context*. The inference component is to give an approximate posterior over the context $q(c|D)$ for a given dataset produced by a *statistic network*.

# 3 NEURAL STATISTICIAN

In order to exploit the assumption of a hierarchical generative process over datasets we will use a 'parameter-transfer approach' (see Pan & Yang, 2010) to extend the variational autoencoder model of Kingma & Welling (2013).

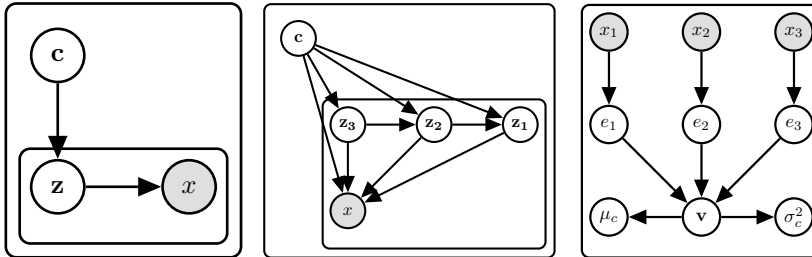

Figure 1: *Left*: basic hierarchical model, where the plate encodes the fact that the context variable $c$ is shared across each item in a given dataset. *Center*: full neural statistician model with three latent layers $z_1, z_2, z_3$. Each collection of incoming edges to a node is implemented as a neural network, the input of which is the concatenation of the edges' sources, the output of which is a parameterization of a distribution over the random variable represented by that node. *Right*: The statistic network, which combines the data via an exchangeable statistic layer.

## 3.1 VARIATIONAL AUTOENCODER

The variational autoencoder is a latent variable model $p(x|z; \theta)$ (often called the decoder) with parameters $\theta$. For each observed $x$, a corresponding latent variable $z$ is drawn from $p(z)$ so that

$$p(x) = \int p(x|z; \theta)p(z) \, dz. \tag{1}$$

The generative parameters $\theta$ are learned by introducing a recognition network (also called an encoder) $q(z|x; \phi)$ with parameters $\phi$. The recognition network gives an approximate posterior over the latent variables that can then be used to give the standard variational lower bound (Saul & Jordan, 1996) on the single-datum log-likelihood. I.e. $\log P(x|\theta) \geq \mathcal{L}_x$, where

$$\mathcal{L}_x = \mathbb{E}_{q(z|x, \phi)} \left[ \log p(x|z; \theta) \right] - D_{KL} \left( q(z|x; \phi) \| p(z) \right). \tag{2}$$

Likewise the full-data log likelihood is lower bounded by the sum of the $\mathcal{L}_x$ terms over the whole dataset. We can then optimize this lower bound with respect to $\phi$ and $\theta$ using the reparameterization trick introduced by Kingma & Welling (2013) and Rezende et al. (2014) to get a Monte-Carlo estimate of the gradient.

## 3.2 BASIC MODEL

We extend the variational autoencoder to the model depicted on the left in Figure 1. This includes a latent variable $c$, the context, that varies between different datasets but is constant, a priori, for items within the same dataset. Now, the likelihood of the parameters $\theta$ for one single particular dataset $D$ is given by

$$p(D) = \int p(c) \left[ \prod_{x \in D} \int p(x|z; \theta)p(z|c; \theta) \, dz \right] dc. \tag{3}$$

The prior $p(c)$ is chosen to be a spherical Gaussian with zero mean and unit variance. The conditional $p(z|c; \theta)$ is Gaussian with diagonal covariance, where all the mean and variance parameters depend on $c$ through a neural network. Similarly the observation model $p(x|z; \theta)$ will be a simple likelihood function appropriate to the data modality with dependence on $z$ parameterized by a neural network. For example, with real valued data, a diagonal Gaussian likelihood could be used where the mean and log variance of $x$ are created from $z$ via a neural network.

We use approximate inference networks $q(z|x, c; \phi)$, $q(c|D; \phi)$, with parameters collected into $\phi$, to once again enable the calculation and optimization of a variational lower bound on the log-likelihood. The single dataset log likelihood lower bound is given by

$$\mathcal{L}_D = \mathbb{E}_{q(c|D;\phi)} \left[ \sum_{x \in d} \mathbb{E}_{q(z|c,x;\phi)} \left[ \log p(x|z;\theta) \right] - D_{KL}\left(q(z|c,x;\phi) \| p(z|c;\theta)\right) \right]$$
$$- D_{KL}\left(q(c|D;\phi) \| p(c)\right). \quad (4)$$

As with the generative distributions, the likelihood forms for $q(z|x, c; \phi)$ and $q(c|D; \phi)$ are diagonal Gaussian distributions, where all the mean and log variance parameters in each distribution are produced by a neural network taking the conditioning variables as inputs. Note that $q(c|D; \phi)$ accepts as input a *dataset $D$* and we refer to this as the statistic network. We describe this in Subsection 3.4.

The full-data variational bound is given by summing the variational bound for each dataset in our collection of datasets. It is by learning the difference of the within-dataset and between-dataset distributions that we are able to discover an appropriate statistic network.

## 3.3 FULL MODEL

The basic model works well for modelling simple datasets, but struggles when the datasets have complex internal structure. To increase the sophistication of the model we use multiple stochastic layers $z_1, \ldots, z_k$ and introduce skip-connections for both the inference and generative networks. The generative model is shown graphically in Figure 1 in the center. The probability of a dataset $D$ is then given by

$$p(D) = \int p(c) \prod_{x \in D} \int p(x|c, z_{1:L}; \theta) p(z_L|c; \theta) \prod_{i=1}^{L-1} p(z_i|z_{i+1}, c; \theta) \, dz_{1:L} \, dc \quad (5)$$

where the $p(z_i|z_{i+1}, c, \theta)$ are again Gaussian distributions where the mean and log variance are given as the output of neural networks. The generative process for the full model is described in Algorithm 1.

The full approximate posterior factorizes analogously as

$$q(c, z_{1:L}|D; \phi) = q(c|D; \phi) \prod_{x \in D} q(z_L|x, c; \phi) \prod_{i=1}^{L-1} q(z_i|z_{i+1}, x, c; \phi). \quad (6)$$

For convenience we give the variational lower bound as sum of a three parts, a reconstruction term $R_D$, a context divergence $C_D$ and a latent divergence $L_D$:

$$\mathcal{L}_D = R_D + C_D + L_D \text{ with} \quad (7)$$

$$R_D = \mathbb{E}_{q(c|D;\phi)} \sum_{x \in D} \mathbb{E}_{q(z_{1:L}|c,x;\phi)} \log p(x|z_{1:L}, c; \theta) \quad (8)$$

$$C_D = D_{KL}\left(q(c|D;\phi) \| p(c)\right) \quad (9)$$

$$L_D = \mathbb{E}_{q(c,z_{1:L}|D;\phi)} \left[ \sum_{x \in D} D_{KL}\left(q(z_L|c,x;\phi) \| p(z_L|c;\theta)\right) \right.$$
$$\left. + \sum_{i=1}^{L-1} D_{KL}\left(q(z_i|z_{i+1}, c, x; \phi) \| p(z_i|z_{i+1}, c; \theta)\right) \right]. \quad (10)$$

The skip-connections $p(z_i|z_{i+1}, c; \theta)$ and $q(z_i|z_{i+1}, x; \phi)$ allow the context to specify a more precise distribution for each latent variable by explaining-away more generic aspects of the dataset at each stochastic layer. This architecture was inspired by recent work on probabilistic ladder networks in Kaae Sønderby et al. (2016). Complementing these are the skip-connections from each latent variable to the observation $p(x|z_{1:L}, c; \theta)$, the intuition here is that each stochastic layer can focus on representing a certain level of abstraction, since its information does not need to be copied into the next layer, a similar approach was used in Maaløe et al. (2016).

Once again, note that we are maximizing the lower bound to the log likelihood over many datasets $D$: we want to maximize the expectation of $\mathcal{L}_D$ over all datasets. We do this optimization using stochastic gradient descent. In contrast to a variational autoencoder where a minibatch would consist of a subsample of *datapoints* from the dataset, we use minibatches consisting of a subsample of *datasets* - tensors of shape (`batch size, sample size, number of features`).

## 3.4 STATISTIC NETWORK

In addition to the standard inference networks we require a *statistic network* $q(c|D;\phi)$ to give an approximate posterior over the context $c$ given a dataset $D = \{x_1, \ldots, x_k\}$. This inference network must capture the exchangeability of the data in $D$.

We use a feedforward neural network consisting of three main elements:

- An instance encoder $E$ that takes each individual datapoint $x_i$ to a vector $e_i = E(x_i)$.
- An exchangeable instance pooling layer that collapses the matrix $(e_1, \ldots, e_k)$ to a single pre-statistic vector $v$. Examples include elementwise means, sums, products, geometric means and maximum. We use the sample mean for all experiments.
- A final post-pooling network that takes $v$ to a parameterization of a diagonal Gaussian.

The graphical model for this is given at the right of Figure 1.

We note that the humble sample mean already gives the statistic network a great deal of representational power due to the fact that the instance encoder can learn a representation where averaging makes sense. For example since the instance encoder can approximate a polynomial on a compact domain, and so can the post-pooling network, a statistic network can approximate any moment of a distribution.

## 4 RELATED WORK

Due to the general nature of the problem considered, our work touches on many different topics which we now attempt to summarize.

**Topic models and graphical models** The form of the graphical model in Figure 1 on the left is equivalent to that of a standard topic model. In contrast to traditional topic models we do not use discrete latent variables, or restrict to discrete data. Work such as that by Ranganath et al. (2014) has extended topic models in various directions, but importantly we use flexible conditional distributions and dependency structures parameterized by deep neural networks. Recent work has explored neural networks for document models (see e.g. Miao et al., 2015) but has been limited to modelling datapoints with little internal structure. Along related lines are 'structured variational autoencoders' (see Johnson et al., 2016), where they treat the general problem of integrating graphical models with variational autoencoders.

**Transfer learning** There is a considerable literature on transfer learning, for a survey see Pan & Yang (2010). There they discuss 'parameter-transfer' approaches whereby parameters or priors are shared across datasets, and our work fits into that paradigm. For examples see Lawrence & Platt (2004) where share they priors between Gaussian processes, and Evgeniou & Pontil (2004) where they take an SVM-like approach to share kernels.

**One-shot Learning** Learning quickly from small amounts of data is a topic of great interest. Lake et al. (2015) use Bayesian program induction for one-shot generation and classification, and Koch (2015) train a Siamese (Chopra et al. (2005)) convolutional network for one-shot image classification. We note the relation to the recent work (Rezende et al., 2016) in which the authors use a *conditional* recurrent variational autoencoder capable of one-shot generalization by taking as extra input a conditioning data point. The important differences here are that we *jointly* model datasets and datapoints and consider datasets of any size. Recent approaches to one-shot classification are matching networks (Vinyals et al., 2016b) (which was concurrent with the initial preprint of this work), and related previous work (Santoro et al., 2016). The former can be considered a kind of differentiable nearest neighbour classifier, and the latter augments their network with memory to store information about the classification problem. Both are trained end-to-end for the classification problem, whereas the present work is a general approach to learning representations of datasets. Probably the closest previous work is by Salakhutdinov et al. (2012) where the authors learn a topic

model over the activations of a DBM for one-shot learning. Compared with their work we use modern architectures and easier to train VAEs, in particular we have fast and amortized feedforward inference for test (and training) datasets, avoiding the need for MCMC.

**Multiple-Instance Learning**  There is previous work on classifying sets in multiple-instance learning, for a useful survey see Cheplygina et al. (2015). Typical approaches involve adapting kernel based methods such as *support measure machines* (Muandet et al., 2012), *support distribution machines* (Póczos et al., 2012) and *multiple-instance-kernels* (Gartner et al., 2002). We do not consider applications to multiple-instance learning type problems here, but it may be fruitful to do so in the future.

**Set2Seq**  In very related work, Vinyals et al. (2016a) explore architectures for mapping sets to sequences. There they use an LSTM to repeatedly compute weighted-averages of the datapoints and use this to tackle problems such as sorting a list of numbers. The main difference between their work and ours is that they primarily consider supervised problems, whereas we present a general unsupervised method for learning representations of sets of i.i.d instances. In future work we may also explore recurrently computing statistics.

**ABC**  There has also been work on learning summary statistics for Approximate Bayesian Computation by either learning to predict the parameters generating a sample as a supervised problem, or by using kernel embeddings as infinite dimensional summary statistics. See the work by Fukumizu et al. (2013) for an example of kernel-based approaches. More recently Jiang et al. (2015) used deep neural networks to predict the parameters generating the data. The crucial differences are that their problem is supervised, they do not leverage any exchangeability properties the data may have, nor can it deal with varying sample sizes.

## 5 EXPERIMENTAL RESULTS

Given an input set $x_1, \ldots x_k$ we can use the statistic network to calculate an approximate posterior over contexts $q(c|x_1, \ldots, x_k; \phi)$. Under the generative model, each context $c$ specifies a conditional model $p(x|c; \theta)$. To get samples from the model corresponding to the most likely posterior value of $c$, we set $c$ to the mean of the approximate posterior and then sample directly from the conditional distributions. This is described in Algorithm 2. We use this process in our experiments to show samples. In all experiments, we use the Adam optimization algorithm (Kingma & Ba, 2014) to optimize the parameters of the generative models and variational approximations. Batch normalization (Ioffe & Szegedy, 2015) is implemented for convolutional layers and we always use a batch size of 16. We primarily use the Theano (Theano Development Team, 2016) framework with the Lasagne (Dieleman et al., 2015) library, but the final experiments with face data were done using Tensorflow (Abadi et al., 2015). In all cases experiments were terminated after a given number of epochs when training appeared to have sufficiently converged (300 epochs for omniglot, youtube and spatial MNIST examples, and 50 epochs for the synthetic experiment).

### 5.1 SIMPLE 1-D DISTRIBUTIONS

In our first experiment we wanted to know if the neural statistician will learn to cluster synthetic 1-D datasets by distribution family. We generated a collection of synthetic 1-D datasets each containing 200 samples. Datasets consist of samples from either an Exponential, Gaussian, Uniform or Laplacian distribution with equal probability. Means and variances are sampled from $U[-1, 1]$ and $U[0.5, 2]$ respectively. The training data contains $10K$ sets.

The architecture for this experiment contains a single stochastic layer with 32 units for $z$ and 3 units for $c$, . The model $p(x|z, c; \theta)$ and variational approximation $q(z|x, c; \phi)$ are each a diagonal Gaussian distribution with all mean and log variance parameters given by a network composed of three dense layers with ReLU activations and 128 units. The statistic network determining the mean and log variance parameters of posterior over context variables is composed of three dense layers before and after pooling, each with 128 units with Rectified Linear Unit (ReLU) activations.

Figure 2 shows 3-D scatter plots of the summary statistics learned. Notice that the different families of distribution cluster. It is interesting to observe that the Exponential cluster is differently orientated to the others, perhaps reflecting the fact that it is the only non-symmetric distribution. We also see that between the Gaussian and Laplacian clusters there is an area of ambiguity which is as one

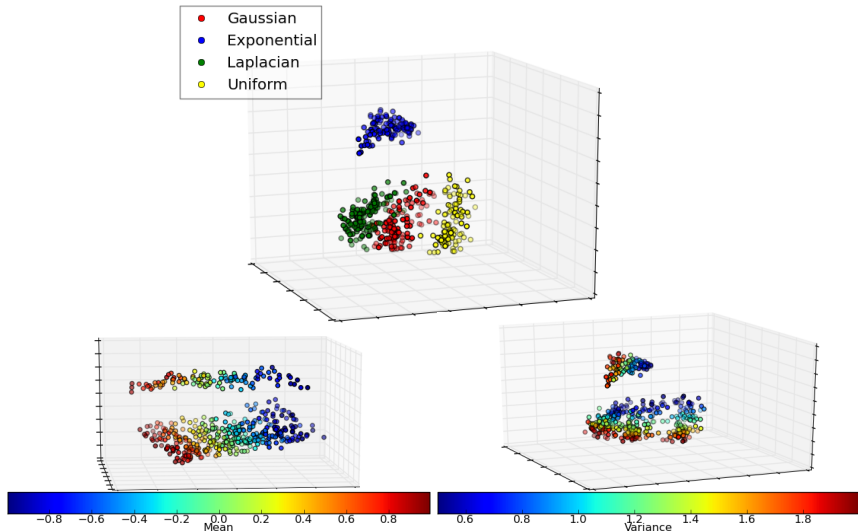

Figure 2: Three different views of the same data. Each point is the mean of the approximate posterior over the context $q(c|D; \phi)$ where $c \in \mathbb{R}^3$. Each point is a summary statistic for a single dataset with 200 samples. Top plot shows points colored by distribution family, left plot colored by the mean and right plot colored by the variance. The plots have been rotated to illustrative angles.

might expect. We also see that within each cluster the mean and variance are mapped to orthogonal directions.

## 5.2 SPATIAL MNIST

Building on the previous experiments we investigate 2-D datasets that have complex structure, but the datapoints contain little information by themselves, making it a good test of the statistic network. We created a dataset called *spatial MNIST*. In spatial MNIST each image from MNIST (LeCun et al., 1998) is turned into a dataset by interpreting the normalized pixel intensities as a probability density and sampling *coordinate values*. An example is shown in Figure 3. This creates two-dimensional spatial datasets. We used a sample size of 50. Note that since the pixel coordinates are discrete, it is necessary to dequantize them by adding uniform noise $u \sim U[0, 1]$ to the coordinates if one models them as real numbers, else you can get arbitrarily high densities (see Theis et al. (2016) for a discussion of this point).

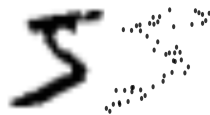

Figure 3: An *image* from MNIST on the left, transformed to a *set* of 50 $(x, y)$ coordinates, shown as a scatter plot on the right.

The generative architecture for this experiment contains 3 stochastic $z$ layers, each with 2 units, and a single $c$ layer with 64 units. The means and log variances of the Gaussian likelihood for $p(x|z_{1:3}, c; \theta)$, and each subnetwork for $z$ in both the encoder and decoder contained 3 dense layers with 256 ReLU units each. The statistic network also contained 3 dense layers pre-pooling and 3 dense layers post pooling with 256 ReLU units.

In addition to being able to sample from the model conditioned on a set of inputs, we can also summarize a dataset by choosing a subset $S \subseteq D$ to minimise the KL divergence of $q(C|D; \phi)$ from $q(C|S; \phi)$. We do this greedily by iteratively discarding points from the full sample. Pseudocode for this process is given in Algorithm 3. The results are shown in Figure 4. We see that the model is capable of handling complex arrangements of datapoints. We also see that it can select sensible subsets of a dataset as a summary.

## 5.3 OMNIGLOT

Next we work with the OMNIGLOT data (Lake et al., 2015). This contains 1628 classes of handwritten characters but with just 20 examples per class. This makes it an excellent test-bed for transfer / few-shot learning. We constructed datasets by splitting each class into datasets of size 5. We train

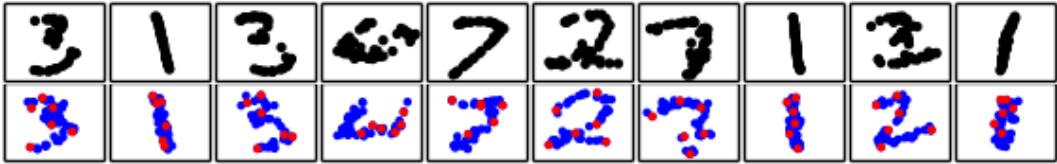

Figure 4: Conditioned samples from spatial MNIST data. Blue and red digits are the input sets, black digits above correspond to samples given the input. Red points correspond to a 6-sample summary of the dataset

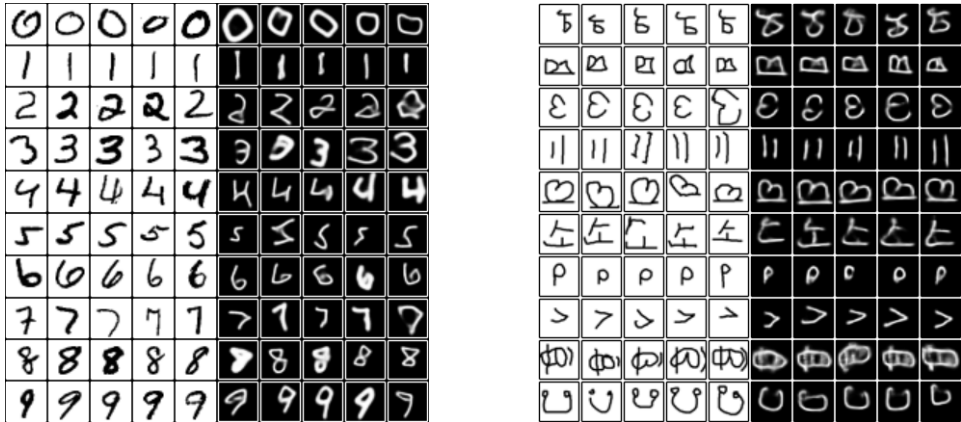

Figure 5: Few-shot learning *Left*: Few-shot learning from OMNIGLOT to MNIST. Left rows are input sets, right rows are samples given the inputs. *Right*: Few-shot learning from with OMNIGLOT data to unseen classes. Left rows are input sets, right rows are samples given the inputs. Black-white inversion is applied for ease of viewing.

on datasets drawn from 1200 classes and reserve the remaining classes to test few-shot sampling and classification. We created new classes by rotating and reflecting characters. We resized the images to $28 \times 28$. We sampled a binarization of each image for each epoch. We also randomly applied the dilation operator from computer vision as further data augmentation since we observed that the stroke widths are quite uniform in the OMNIGLOT data, whereas there is substantial variation in MNIST, this augmentation improved the visual quality of the few-shot MNIST samples considerably and increased the few-shot classification accuracy by about 3 percent. Finally we used 'sample dropout' whereby a random subset of each dataset was removed from the pooling in the statistic network, and then included the number of samples remaining as an extra feature. This was beneficial since it reduced overfitting and also allowed the statistic network to learn to adjust the approximate posterior over $c$ based on the number of samples.

We used a single stochastic layer with 16 units for $z$, and 512 units for $c$. We used a shared convolutional encoder between the inference and statistic networks and a deconvolutional decoder network. Full details of the networks are given in Appendix B.1. The decoder used a Bernoulli likelihood.

In Figure 5 we show two examples of few-shot learning by conditioning on samples of *unseen* characters from OMNIGLOT, and conditioning on samples of digits from MNIST. The samples are mostly of a high-quality, and this shows that the neural statistician can generalize even to new datasets.

As a further test we considered few-shot classification of both unseen OMNIGLOT characters and MNIST digits. Given a sets of labelled examples of each class $D_0, \ldots, D_9$ (for MNIST say), we computed the approximate posteriors $q(C|D_i; \phi)$ using the statistic network. Then for each test image $x$ we also computed the posterior $q(C|x; \phi)$ and classified it according to the training dataset $D_i$ minimizing the KL divergence *from* the test context *to* the training context. This process is described in Algorithm 4. We tried this with either 1 or 5 labelled examples per class and either 5 or 20 classes. For each trial we randomly select $K$ classes, randomly select training examples for each class, and test on the remaining examples. This process is repeated 100 times and the results averaged. The results are shown in Table 1. We compare to a number of results reported in Vinyals et al. (2016b) including Santoro et al. (2016) and Koch (2015). Overall we see that

the neural statistician model can be used as a strong classifier, particularly for the 5-way tasks, but performs worse than matching networks for the 20-way tasks. One important advantage that matching networks have is that, whilst each class is processed independently in our model, the representation in matching networks is conditioned on all of the classes in the few-shot problem. This means that it can exaggerate differences between similar classes, which are more likely to appear in a 20-way problem than a 5-way problem.

| | Task | | Method | | | |
|---|---|---|---|---|---|---|
| Test Dataset | K Shot | K Way | Siamese | MANN | Matching | Ours |
| MNIST | 1 | 10 | 70 | - | 72 | **78.6** |
| MNIST | 5 | 10 | - | - | - | 93.2 |
| OMNIGLOT | 1 | 5 | 97.3 | 82.8 | **98.1** | **98.1** |
| OMNIGLOT | 5 | 5 | 98.4 | 94.9 | 98.9 | **99.5** |
| OMNIGLOT | 1 | 20 | 88.1 | - | **93.8** | 93.2 |
| OMNIGLOT | 5 | 20 | 97.0 | - | **98.7** | 98.1 |

Table 1: The table shows the classification accuracies of various few-shot learning tasks. Models are trained on OMNIGLOT data and tested on either unseen OMNIGLOT classes or MNIST with varying numbers of samples per class (K-shot) with varying numbers of classes (K-way). Comparisons are to Vinyals et al. (2016b) (Matching), Santoro et al. (2016) (MANN) and Koch (2015) (Siamese). 5-shot MNIST results are included for completeness.

## 5.4 YOUTUBE FACES

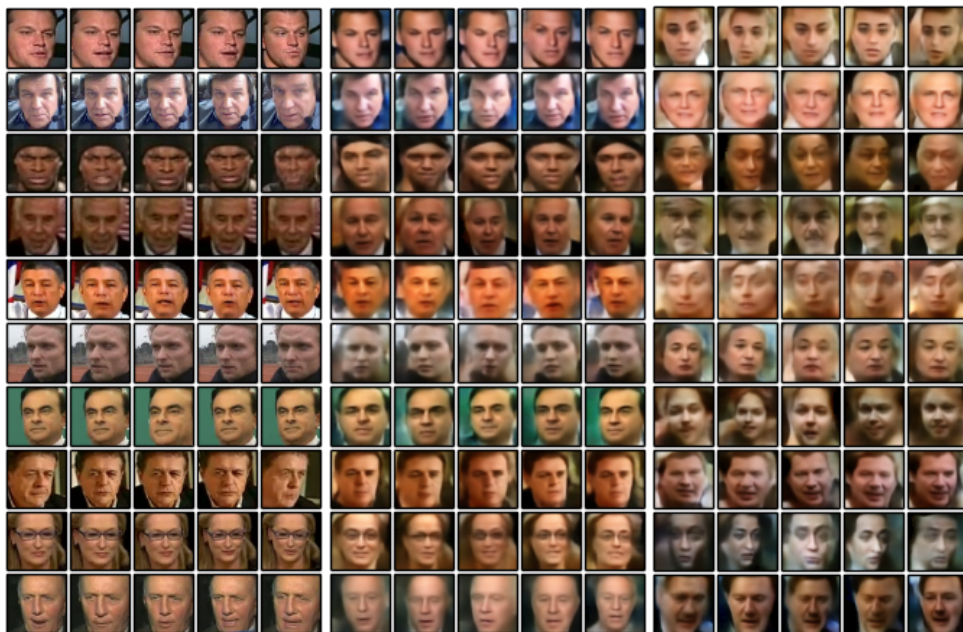

Figure 6: Few-shot learning for face data. Samples are from model trained on Youtube Faces Database. *Left*: Each row shows an input set of size 5. *Center*: Each row shows 5 samples from the model corresponding to the input set on the left. *Right*: Imagined new faces generated by sampling contexts from the prior. Each row consists of 5 samples from the model given a particular sampled context.

Finally, we provide a proof of concept for generating faces of a particular person. We use the Youtube Faces Database from Wolf et al. (2011). It contains $3,245$ videos of $1,595$ different people. We use the aligned and cropped to face version, resized to $64 \times 64$. The validation and test sets contain 100 unique people each, and there is no overlap of persons between data splits. The sets were created by sampling frames randomly without replacement from each video, we use a set size of 5 frames. We resample the sets for the training data each epoch.

Our architecture for this problem is based on one presented in Lamb et al. (2016). We used a single stochastic layer with $500$ dimensional latent $c$ and 16 dimensional $z$ variable. The statistic network and the inference network $q(z|x,c;\phi)$ share a common convolutional encoder, and the deocder uses deconvolutional layers. For full details see Appendix B.2. The likelihood function is a Gaussian, but where the variance parameters are shared across all datapoints, this was found to make training faster and more stable.

The results are shown in Figure 6. Whilst there is room for improvement, we see that it is possible to specify a complex distribution on-the-fly with a set of photos of a previously unseen person. The samples conditioned on an input set have a reasonable likeness of the input faces. We also show the ability of the model to generate new datasets and see that the samples have a consistent identity and varied poses.

## 6 CONCLUSION

We have demonstrated a highly flexible model on a variety of tasks. Going forward our approach will naturally benefit from advances in generative models as we can simply upgrade our base generative model, and so future work will pursue this. Compared with some other approaches in the literature for few-shot learning, our requirement for supervision is weaker: we only ask at training time that we are given datasets, but we do not need labels for the datasets, nor even information on whether two datasets represent the same or different classes. It would be interesting then to explore application areas where only this weaker form of supervision is available. There are two important limitations to this work, firstly that the method is dataset hungry: it will likely not learn useful representations of datasets given only a small number of them. Secondly at test time the few-shot fit of the generative model will not be greatly improved by using larger datasets unless the model was also trained on similarly large datasets. The latter limitation seems like a promising future research direction - bridging the gap between fast adaptation and slow training.

### ACKNOWLEDGMENTS

This work was supported in part by the EPSRC Centre for Doctoral Training in Data Science, funded by the UK Engineering and Physical Sciences Research Council (grant EP/L016427/1) and the University of Edinburgh.

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

## A    APPENDIX A: PSEUDOCODE

---
**Algorithm 1** Sampling a dataset of size $k$

---
sample $c \sim p(c)$
**for** $i = 1$ **to** $k$ **do**
    sample $z_{i,L} \sim p(z_L|c; \theta)$
    **for** $j = L - 1$ **to** 1 **do**
        sample $z_{i,j} \sim p(z_j|z_{i,j+1}, c; \theta)$
    **end for**
    sample $x_i \sim p(x|z_{i,1}, \ldots, z_{i,L}, c; \theta)$
**end for**

---

---
**Algorithm 2** Sampling a dataset of size $k$ conditioned on a dataset of size $m$

---
$\mu_c, \sigma_c^2 \leftarrow q(c|x_1, \ldots, x_m; \phi)$ {Calculate approximate posterior over $c$ using statistic network.}
$c \leftarrow \mu_c$ {Set $c$ to be the mean of the approximate posterior.}
**for** $i = 1$ **to** $k$ **do**
    sample $z_{i,L} \sim p(z_L|c; \theta)$
    **for** $j = L - 1$ **to** 1 **do**
        sample $z_{i,j} \sim p(z_j|z_{i,j+1}, c; \theta)$
    **end for**
    sample $x_i \sim p(x|z_{i,1}, \ldots, z_{i,L}, c; \theta)$
**end for**

---

---
**Algorithm 3** Selecting a representative sample of size $k$

---
$S \leftarrow \{x_1, \ldots, x_m\}$
$I \leftarrow \{1, \ldots, m\}$
$S_I = \{x_i \in S : i \in I\}$
$N_{S_I} \leftarrow q(c|S_I; \phi)$ {Calculate approximate posterior over $c$ using statistic network.}
**for** $i = 1$ **to** $k$ **do**
    $t \leftarrow argmin_{j \in I} D_{KL}\left(N_S \| N_{S_{I-j}}\right)$
    $I \leftarrow I - t$
**end for**

---

---

**Algorithm 4** $K$-way few-shot classification

---

$D_0, \ldots, D_K \leftarrow$ sets of labelled examples for each class
$x \leftarrow$ datapoint to be classified
$N_x \leftarrow q(c|x; \phi)$ {approximate posterior over $c$ given query point}
**for** $i = 1$ **to** $K$ **do**
    $N_i \leftarrow q(c|D_i; \phi)$
**end for**
$\hat{y} \leftarrow argmin_i D_{KL}\left(N_i \| N_x\right)$

---

## B    APPENDIX B: FURTHER EXPERIMENTAL DETAILS

### B.1    OMNIGLOT

**Shared encoder** $x \to h$

---

$2\times \{$ *conv2d* 64 feature maps with $3 \times 3$ kernels and ELU activations $\}$
*conv2d* 64 feature maps with $3 \times 3$ kernels, stride 2 and ELU activations
$2\times \{$*conv2d* 128 feature maps with $3 \times 3$ kernels and ELU activations $\}$
*conv2d* 128 feature maps with $3 \times 3$ kernels, stride 2 and ELU activations
$2\times \{$ *conv2d* 256 feature maps with $3 \times 3$ kernels and ELU activations $\}$
*conv2d* 256 feature maps with $3 \times 3$ kernels, stride 2 and ELU activations

**Statistic network** $q(c|D; \phi) : h_1, \ldots, h_k \to \mu_c, \sigma_c^2$

---

*fully-connected* layer with 256 units and ELU activations
*sample-dropout* and *concatenation* with number of samples
*average pooling* within each dataset
$2\times \{$*fully-connected* layer with 256 units and ELU activations $\}$
*fully-connected* linear layers to $\mu_c$ and $\log \sigma_c^2$

**Inference network** $q(z|x, c; \phi) : h, c \to \mu_z, \sigma_z^2$

---

*concatenate* $c$ and $h$
$3\times \{$*fully-connected* layer with 256 units and ELU activations $\}$
*fully-connected* linear layers to $\mu_z$ and $\log \sigma_z^2$

**Latent decoder network** $p(z|c; \theta) : c \to \mu_z, \sigma_z^2$

---

$3\times \{$*fully-connected* layer with 256 units and ELU activations $\}$
*fully-connected* linear layers to $\mu_z$ and $\log \sigma_z^2$

**Observation decoder network** $p(x|c, z; \theta) : c, z \to \mu_x$

---

*concatenate* $z$ and $c$
*fully-connected* linear layers with $4 \cdot 4 \cdot 256$ units
$2\times \{$ *conv2d* 256 feature maps with $3 \times 3$ kernels and ELU activations $\}$
*deconv2d* 256 feature maps with $2 \times 2$ kernels, stride 2, ELU activations
$2\times \{$ *conv2d* 128 feature maps with $3 \times 3$ kernels and ELU activations $\}$
*deconv2d* 128 feature maps with $2 \times 2$ kernels, stride 2, ELU activations
$2\times \{$ *conv2d* 64 feature maps with $3 \times 3$ kernels and ELU activations $\}$
*deconv2d* 64 feature maps with $2 \times 2$ kernels, stride 2, ELU activations
*conv2d* 1 feature map with $1 \times 1$ kernels, sigmoid activations

## B.2 YOUTUBE FACES

**Shared encoder** $x \rightarrow h$

$2\times$ { *conv2d* 32 feature maps with $3 \times 3$ kernels and ELU activations }
*conv2d* 32 feature maps with $3 \times 3$ kernels, stride 2 and ELU activations
$2\times$ {*conv2d* 64 feature maps with $3 \times 3$ kernels and ELU activations }
*conv2d* 64 feature maps with $3 \times 3$ kernels, stride 2 and ELU activations
$2\times$ { *conv2d* 128 feature maps with $3 \times 3$ kernels and ELU activations }
*conv2d* 128 feature maps with $3 \times 3$ kernels, stride 2 and ELU activations
$2\times$ { *conv2d* 256 feature maps with $3 \times 3$ kernels and ELU activations }
*conv2d* 256 feature maps with $3 \times 3$ kernels, stride 2 and ELU activations

**Statistic network** $q(c|D, \phi) : h_1, \ldots, h_k \rightarrow \mu_c, \sigma_c^2$

*fully-connected* layer with 1000 units and ELU activations
*average pooling* within each dataset
*fully-connected* linear layers to $\mu_c$ and $\log \sigma_c^2$

**Inference network** $q(z|x, c, \phi) : h, c \rightarrow \mu_z, \sigma_z^2$

*concatenate* $c$ and $h$
*fully-connected* layer with 1000 units and ELU activations
*fully-connected* linear layers to $\mu_z$ and $\log \sigma_z^2$

**Latent decoder network** $p(z|c, ; \theta) : c \rightarrow \mu_z, \sigma_z^2$

*fully-connected* layer with 1000 units and ELU activations
*fully-connected* linear layers to $\mu_z$ and $\log \sigma_z^2$

**Observation decoder network** $p(x|c, z; \theta) : c, z \rightarrow \mu_x$

*concatenate* $z$ and $c$
*fully-connected* layer with 1000 units and ELU activations
*fully-connected* linear layer with $8 \cdot 8 \cdot 256$ units
$2\times$ { *conv2d* 256 feature maps with $3 \times 3$ kernels and ELU activations }
*deconv2d* 256 feature maps with $2 \times 2$ kernels, stride 2, ELU activations
$2\times$ { *conv2d* 128 feature maps with $3 \times 3$ kernels and ELU activations }
*deconv2d* 128 feature maps with $2 \times 2$ kernels, stride 2, ELU activations
$2\times$ { *conv2d* 64 feature maps with $3 \times 3$ kernels and ELU activations }
*deconv2d* 64 feature maps with $2 \times 2$ kernels, stride 2, ELU activations
$2\times$ { *conv2d* 32 feature maps with $3 \times 3$ kernels and ELU activations }
*deconv2d* 32 feature maps with $2 \times 2$ kernels, stride 2, ELU activations
*conv2d* 3 feature maps with $1 \times 1$ kernels, sigmoid activations

