# Peer review of "Towards a Neural Statistician"

_ICLR 2017 — accepted_

[Official Review · AnonReviewer3 · rating 6 · confidence 4 · 16 Dec 2016]
**Interesting paper that starts to expand the repertoire of variational autoencoders**
clarity 3

The authors introduce a variant of the variational autoencoder (VAE) that models dataset-level latent variables. The idea is clearly motivated and well described. In my mind the greatest contribution of this paper is the movement beyond the relatively simple graphical model structure of the traditional VAEs and the introduction of more interesting structures to the deep learning community. 

Comments:

- It's not clear to me why this should be called a "statistician". Learning an approximate posterior over summary statistics is not the only imaginable way to summarize a dataset with a neural network. One could consider a maximum likelihood approach, etc. In general it felt like the paper could be more clear, if it avoided coining new terms like "statistic network" and stuck to the more accurate "approximate posterior".

- The experiments are nice, and I appreciate the response to my question regarding "one shot generation". I still think that language needs to be clarified, specifically at the end of page 6. My understanding of Figure 5 is the following: Take an input set, compute the approximate posterior over the context vector, then generate from the forward model given samples from the approximate posterior. I would like clarification on the following: 

(a) Are the data point dependent vectors z generated from the forward model or taken from the approximate posterior? 

(b) I agree that the samples are of high-quality, but that is not a quantified statement. The advantage of VAEs over GANs is that we have natural ways of computing log-probabilities. To that end, one "proper" way of computing the "one shot generation" performance is to report log p(x | c) (where c is sampled from the approximate posterior) or log p(x) for held-out datasets. I suspect that log probability performance of these networks relative to a vanilla VAE without the context latent variable will be impressive. I still don't see a reason not to include that.

[Official Review · AnonReviewer4 · rating 8 · confidence 2 · 19 Dec 2016]
**Solid Contribution**

This paper proposes a hierarchical generative model where the lower level consists of points within datasets and the higher level models unordered sets of datasets.  The basic idea is to use a "double" variational bound where a higher level latent variable describes datasets and a lower level latent variable describes individual examples.  

Hierarchical modeling is an important and high impact problem, and I think that it's under-explored in the Deep Learning literature.  

Pros:
  -The few-shot learning results look good, but I'm not an expert in this area.  
  -The idea of using a "double" variational bound in a hierarchical generative model is well presented and seems widely applicable.  

Questions: 
  -When training the statistic network, are minibatches (i.e. subsets of the examples) used?  
  -If not, does using minibatches actually give you an unbiased estimator of the full gradient (if you had used all examples)?  For example, what if the statistic network wants to pull out if *any* example from the dataset has a certain feature and treat that as the characterization.  This seems to fit the graphical model on the right side of figure 1.  If your statistic network is trained on minibatches, it won't be able to learn this characterization, because a given minibatch will be missing some of the examples from the dataset.  Using minibatches (as opposed to using all examples in the dataset) to train the statistic network seems like it would limit the expressive power of the model.  

Suggestions: 
  -Hierarchical forecasting (electricity / sales) could be an interesting and practical use case for this type of model.

[Official Review · AnonReviewer1 · rating 8 · confidence 4 · 19 Dec 2016]
**a nice addition to the one-/few-shot learning literature**
originality 2 · impact 3 · substance 3 · recommendation (unofficial) 3

Sorry for the late review -- I've been having technical problems with OpenReview which prevented me from posting.

This paper presents a method for learning to predict things from sets of data points. The method is a hierarchical version of the VAE, where the top layer consists of an abstract context unit that summarizes a dataset. Experiments show that the method is able to "learn to learn" by acquiring the ability to learn distributions from small numbers of examples.

Overall, this paper is a nice addition to the literature on one- or few-shot learning. The method is conceptually simple and elegant, and seems to perform well. Compared to other recent papers on one-shot learning, the proposed method is simpler, and is based on unsupervised representation learning. The paper is clearly written and a pleasure to read.

The name of the paper is overly grandiose relative to what was done; the proposed method doesn’t seem to have much in common with a statistician, unless one means by that "someone who thinks up statistics". 

The experiments are well chosen, and the few-shot learning results seem pretty solid given the simplicity of the method.

The spatial MNIST dataset is interesting and might make a good toy benchmark. The inputs in Figure 4 seem pretty dense, though; shouldn’t the method be able to recognize the distribution with fewer samples?  (Nitpick: the red points in Figure 4 don’t seem to correspond to meaningful points as was claimed in the text.) 

Will the authors release the code?

[Final Decision · Program Chairs · 06 Feb 2017]
**ICLR committee final decision**

This is an interesting paper that adds nicely to the literature on VAEs and one-shot generalisation. This will be of interest to the community and will contribute positively to the conference.